# Enjoyment of Physical Activity—Not MVPA during Physical Education—Predicts Future MVPA Participation and Sport Self-Concept

**DOI:** 10.3390/sports9090128

**Published:** 2021-09-10

**Authors:** Jared D. Ramer, Natalie E. Houser, Robert J. Duncan, Eduardo E. Bustamante

**Affiliations:** 1College of Applied Health Sciences, University of Illinois at Chicago, Chicago, IL 60607, USA; ebusta2@uic.edu; 2College of Kinesiology, University of Saskatchewan, Saskatoon, SK S7N 5B2, Canada; natalie.houser@usask.ca; 3College of Health and Human Sciences, Purdue University, West Lafayette, IN 47907, USA; duncan99@purdue.edu

**Keywords:** physical literacy, physical exercise, development, exercise behavior, children, school

## Abstract

There exists a general understanding that physical education (PE) is a means to create a physically healthy population. However, disagreements arise over primary practices within PE to achieve this end. The primary divergence is whether PE facilitators should primarily ensure participants exert a specific level of energy during class or develop their confidence, competence, knowledge, and motivation for meaningful physical activity (PA) participation outside of the PE classroom (referred to as physical literacy (PL)). This study uses structural equation modeling to examine associations between enjoyment of PA and minutes of moderate to vigorous physical activity (MVPA) in PE class in grade 5 (mean age = 10) and participation in PA and feelings about PA 1 year later, in grade 6 (mean age = 11), in the NICHD Study of Early Child Care and Youth Development (SECCYD, *N* = 1364). Enjoyment of PA in grade 5 predicted sport self-concept (*β* = 0.347, *p* ≤ 0.001), MVPA (*β* = 0.12, *p* ≤ 0.001), and enjoyment of PA (*β* = 0.538, *p* ≤ 0.001) in grade 6. These associations remained when including weekday MVPA performed in grade 5 as an indirect effect (β = 0.058, *p* ≤ 0.001). MVPA performed during PE in grade 5 was not associated with any PA outcomes in grade 6. Findings suggest PE instructors should prioritize PL development over maintenance of high energy expenditure during PE class for long-term MVPA.

## 1. Introduction

Though exercise and physical activity (PA) are very similar, the two terms are not synonymous. PA is any bodily movement produced by skeletal muscles that increases energy expenditure above resting levels; exercise is a subcategory of PA with the components of planning, structure, repetitiveness, and purposive striving toward the objective of physical fitness. Both exercise and PA are often measured in kilocalories and generate improvements in physical fitness as PA intensity, duration, and frequency increase [1]. Studies testing the effects of MVPA on children have demonstrated profound health benefits from participation, including improved adiposity, triglyceride levels, HDL-C levels, inflammation, endothelial function, cardiovascular fitness, musculoskeletal health, insulin sensitivity, brain structure, brain function, cognition, math performance, mental health, and self-concept [2,3,4,5].

Given the stunning list of benefits and the unwavering childhood obesity epidemic, which drove obesity rates from just under 5% in 1963 to roughly 15% today [6], it is understandable that adults have sought ways to get children to participate in MVPA. One means of getting children active is physical education (PE), and in recent years, there has been a push to ensure that children achieve the recommended amount of MVPA during PE class time in order to ensure they remain healthy as they grow and develop. The purpose of this study is to explore whether this approach, designing PE to prioritize minutes of MVPA, is strong for ensuring children maintain high levels of MVPA in the future and to test which characteristics of PA experience determine future MVPA participation.

For children, schools are a cultural touchpoint that provides various opportunities for movement (e.g., physical education, recess) to all who attend. PE, however, is the opportunity most consistently relevant for MVPA. School policies for PE standardization in the U.S., like all curriculum standards, are established at the state level; there are no federal policies for any curriculum standard. In 2010, 43 states required PE in elementary schools, 40 in middle schools, and 46 in high schools [7]. State-level policies in the U.S. generally specified minutes of PE class time and/or percent of PE class time spent in MVPA [8,9]. Even with these policies in place, low levels of monitoring and evaluation make it difficult to ensure objectives are met [9]. However, high school students with specific PE requirements reported an average of 31 additional minutes per week spent physically active during PE classes, and girls were more likely to report a higher number of days per week having exercised vigorously or having engaged in strength-building activities [8].

Previous research using data from the NICHD longitudinal Study of Early Child Care and Youth Development (SECCYD) data has examined MVPA patterns, BMI, and PE quality in children and youth. Findings showed that between the ages of 9 and 15 years, MVPA started at approximately 3 h per day at age 9 and decreased by 38–41 min per year, ending at approximately 49 min per weekday (35 min per weekend day) at age 15. There were also significant sex disparities, as boys were consistently more active than girls [10]. During the same time period (9–15 years of age), there was a negative association between MVPA and change in BMI at the upper tail of the BMI distribution [11]. Nesbit et al. [12] identified BMI trajectories based on demographic characteristics and clusters of obesogenic behaviors between grades 5 and 8. Two trajectories were identified and compared: overweight-obese with BMI ≥ 85th percentile, showing significantly less PA, and no difference in sedentary behavior from healthy weight BMI between 5th and 84th percentile [12]. Finally, third grade PE was examined in this group using the System for Observing Fitness Time (SOFIT) observation method Boys were found to spend proportionately more time in very active and MVPA during PE than girls, and children received a mean of 25 min per week of MVPA in school PE [13]. Nader made the point that this amount of MVPA is under the national recommendations for this age group (60 min MVPA/day); however, it is important to note MVPA recommendations are based on self-report data, and this is not comparable to physical activity monitor and direct observation measures.

Beyond minutes of PA during the school day, PE has other effects (explicit or implicit) on personal and physical development related to students’ participation in future PA. Proximal goals include the development of fundamental movement skills (FMS) expected to translate to participation in sports or other PA pursuits throughout the lifespan. This is evidenced by cross-sectional findings of children with greater competence in FMS having significantly less sedentary time [14] and more time spent in MVPA [15]. Longitudinally, these findings do not hold up as strongly, with one research group finding only object control skills influencing PA [16] mediated by perceived sports competence [17] and another finding no associations [18]. Beyond FMS lies physical literacy (PL), a multidimensional construct transcending FMS to include cognitive, emotional, and social elements that collectively influence movement experiences existentially [19,20]. Each of these constructs—cognitive, emotional, and social—theoretically affect the full experience of PA, such as enjoyment. As a general tenet of intrinsic motivation according to self-determination theory, children who enjoy a variety of different activities will be more likely to be physically active throughout their lives [21,22].

During PE classes, not only do teachers impose a unified goal orientation, but there also exist goal orientations within each individual student. Research has been conducted on goal development and orientation of children within PE contexts through the lens of goal achievement profiles (i.e., mastery, performance-approach, performance-avoidance). One study showed that children endorsing consistently high mastery regardless of performance orientation had the most adaptive motivational responses to future PA. Furthermore, children participating in PE with a consistently high mastery/low performance climate showed decreased performance-avoidance (concerned with avoiding failure in front of others) [23]. This is important, as mastery and performance orientation affect self-concept in all academic subjects. Generally, mastery orientation is related to high self-efficacy [24] and also relates to self-concept. Higher self-concept has been shown to affect motivation to learn, improve, and participate in the subject (e.g., math, reading, sports) in the future [25,26], including physical activity [27] and exercise [28]. Two potential downsides of enforcing high levels of MVPA on all students in a class is that it (1) may take away time opportunities for developing PL skills that will help children remain active in the future, of their own volition, once the structure of daily PE is removed and (2) may create negative associations with both the PE class and PA itself when a student is dehumanized to a series of mechanistic systems of which one learns they must force or be forced to exert energy for obesity metabolism without understood purpose beyond the system itself.

Using the previously mentioned SECCYD dataset, the purpose of this was to examine the associations between PA experiences in grade 5 (i.e., enjoyment of PA, minutes of MVPA in PE class, and weekday MVPA) on psychosocial determinants of PA (i.e., sport self-concept and enjoyment of PA) and health (total MVPA and BMI) in grade 6. Furthermore, minutes of weekday MVPA in grade 5 were used to test as an indirect effect determining whether any direct associations between the independent and dependent variables could be the result of the MVPA minutes attained in grade 5. Control variables included sex, grade 5 BMI, and limited physical functioning. 

Three hypotheses were tested in this model. Hypothesis 1: We hypothesized both independent variables would show small to moderate direct associations with each dependent variable. Hypothesis 2: We also expected both enjoyment of PA and PE MVPA to show indirect effects on grade 6 outcomes through weekday MVPA minutes in grade 5. We hypothesized there would be small to moderate indirect associations between independent to dependent variables through the path of weekday minutes of MVPA in grade 5. Hypothesis 3: We expected the unexplained variance of grade 6 sport self-concept to be related to PA enjoyment and MVPA all days. We hypothesized there will be a small to moderate significant covariance between the residual variances of sport self-concept and PA enjoyment in grade 5 and sport self-concept and MVPA all days in grade 6.

## 2. Materials and Methods

### 2.1. Participants

Families who participated in this study were recruited at the time of their child’s birth and were followed throughout high school and beyond. Recruitment took place during the year 1991 in locations in the following cities: Little Rock, AK; Irvine, CA; Lawrence, KS; Boston, MA; Philadelphia, PA; Pittsburgh, PA; Charlottesville, VA; Morgantown, NC; Seattle, WA; and Madison, WS. When the child aged 1 month, families with healthy babies were enrolled (*N* = 1364 families). Families planning to move or if their newborn remained in the hospital more than seven days were ineligible. The demographics were similar to those of national norms [11]: children were 24% racial/ethnic minority, 14% of mothers were single parents, and 11% had less than high school education. Families who were recruited did not differ significantly on major demographic variables than those families who were not [13]. 

### 2.2. Variables

This study includes measures from grade 5 and grade 6 reported by parents, self, observation, and device-assessed measures. Descriptions of reporting sources and methods of collection are included in Table 1.

### 2.3. Analysis Plan

The first step of creating a structural equation model (SEM) is the creation of a correlation matrix of all study variables. The second is the creation of the hypothetical model based on expected relationships.

Grade 5 and 6 Physical Activity Monitor (PAM) attrition from the full sample was 37.7% and 49%, respectively. Height and weight attrition was 38.8% in grade 5 and 40.6% in grade 6. Remaining attrition ranged from 35.9% to 25.7% barring sex, which had no missing data, having been measured at birth with no reported change during the study. We address these missing data by using full information maximum likelihood (FIML) estimation in which missing values on manifesting variables are used to estimate missing data. The likelihood function is computed for cases with complete data on some variables and other cases with complete data on all variables. The two likelihoods are then maximized to find the estimates. SPSS AMOS was used to compute the FIML estimation as well as the SEM analysis. One advantage of using SEM is that it provides capability for full information likelihood estimation, allowing for all data from all variables in the analysis to be used in informing model estimates. A second advantage is it allows for the testing of the overall model structure, accounting for error variance parameters of both independent and dependent variables, and to compare theoretical differences between contributions from independent variables.

Independent variables for the model were all measured in grade 5 and include enjoys PA, minutes of MVPA in PE, and MVPA during weekdays. Control variables, also from grade 5, include BMI, sex, and limited physical functioning. Dependent variables for the model were all measured in grade 6 and include sport self-concept, enjoys PA, MVPA all days, and BMI. Figure 1 is a visual representation of all associations tested within the SEM.

**Hypothesis** **1** **(H1).**
*Direct effects of independent variables on dependent variables.*


The SEM examined associations separately between grade 5 enjoys PA and PE MVPA on grade 6 outcomes—sport self-concept, enjoys PA, MVPA all days, and BMI. Included grade 5 control variables were sex, BMI, and limited physical functioning.

Independent variables were connected to dependent variables by regression paths.

**Hypothesis** **2** **(H2).**
*Indirect effects of independent variables on dependent variables through the mediator variable.*


Indirect associations were tested by specifying regression paths from the independent variables to grade 5 weekday MVPA and from grade 5 weekday MVPA to all outcome variables.

**Hypothesis** **3** **(H3).**
*Residual variance covariance associations between specific dependent variables.*


Finally, residual error variances of grade 6 variables, including sport self-concept and enjoys PA and sport self-concept and MVPA all days, covariation relationships were tested. This allowed us to observe whether the unexplained variances of these outcome variables are related.

## 3. Results

### 3.1. Descriptive Statistics and Correlations among Variables

Table 2 provides completed measure counts, means, standard deviations, and ranges for all variables. Descriptive statistics were calculated using SPSS.

The Pearson correlation matrix between all study variables is provided as Appendix A. For a complete list of all factor estimates, and direct, indirect, and total effects see Appendix A.

### 3.2. Model Fit

Chi-square score for the model is 7.886 with 4 degrees of freedom. Degrees of freedom in SEM is calculated based on the number of available observations in the empty model correlation matrix minus the number of observations used to estimate parameters. The *p*-value for the difference between the default and independent model is 0.096, with the null hypothesis being the observed data and predicted model are equal; a non-significant result means the theoretical model matrix better predicts the non-parameter matrix (predicted model). This model fit index is not commonly used, as with large sample sizes, there is greater likelihood of finding a significant chi-square statistic; what is more commonly used is the equation χ^2^/df, with good fit being less than 5. This equation resulted in 1.97 for this model, which further indicates this a strong-fitting model.

Root mean square error of approximation (RMSEA), a measure of how far the hypothesized model is from a perfect model and considered strong if less than 0.08, equaled 0.027. The comparative fit index (CFI) and Tucker–Lewis Index (TLI) are both incremental fit indices, with numbers closer to 1 and greater than 0.95, indicating better fit. CFI equaled 0.998, and TLI equaled 0.979. Each of these most commonly used indices show this model has strong model.

### 3.3. Estimates

Figure 2 shows all significant associations for the model.

Hypothesis 1:Direct effects of independent variables on dependent variables.

MVPA minutes during PE per week in grade 5 were not significantly associated with any grade 6 outcomes. Children who enjoyed PA in grade 5 were significantly more likely to have higher sport self-concept (*β* = 0.347, *p* ≤ 0.001), attain more minutes of MVPA all days (*β* = 0.12, *p* ≤ 0.001), and enjoy PA in grade 6 (*β* = 0.538, *p* ≤ 0.001).

Hypothesis 2:Indirect effects of independent variables on dependent variables through the mediator variable.

Weekday MVPA in grade 5 was significantly associated with attaining MVPA all days in grade 6 (*β* = 0.338, *p* ≤ 0.001) but not with any other outcomes. PE MVPA had no association with weekday MVPA in grade 5; however, grade 5 enjoys PA was significantly associated with weekday MVPA (*β* = 0.173) in grade 5.

Indirect effects for PE MVPA through weekday MVPA in grade 5 were strongest for grade 6 MVPA all days (*β* = 0.015). Grade 5 enjoys PA through the same variable (weekday MVPA) showed a stronger association on the same dependent variable (MVPA all days) (*β* = 0.058). Grade 5 BMI (*β* = 0.044) and sex (*β* = 0.076) both showed similar indirect effects on grade 6 MVPA all days. All remaining indirect effects were *β* < 0.01.

Hypothesis 3:Residual variance covariance between specific dependent variables.

The residual (unexplained) variances of grade 6 sport self-concept significantly correlated with the residual variance of both grade 6 enjoys PA (*r* = 0.229, *β* = 0.292, *p* ≤ 0.001) and MVPA all days (*r* = 3.566, *β* = 0.096, *p* = 0.009).

In terms of the explained variance, squared multiple correlations show the variance explained by the predictor variables. Grade 6 BMI was strongly explained by relationship estimations of the model at 90%. Grade 6 MVPA and enjoy PA had a small-moderate variance, explained at 32% and 23%, respectively. Sport self-concept and weekday MVPA had a small amount of variance, explained at 13.3% and 10.3%, respectively.

Relationships with Control Variables:

Sex was associated with PE MVPA (*r* = −1.07, *β* = 0.097, *p* = 0.004) and was not correlated with PA enjoyment. Sex had a small association with weekday MVPA (*β* = 0.223) and MVPA all days (*β* = 0.209) as well as with sport self-concept (*β* = 0.081). Limited physical functioning was not significantly associated with any other variables.

BMI at grade 5 correlated with grade 5 Enjoys PA (*r* = −0.51, *β* = 0.138, *p* ≤ 0.001) and grade 5 PE MVPA (*r* = −9.071, *β* = 0.092, *p* = 0.012). BMI at grade 5 had a small relationship with grade 5 weekday MVPA (*β* = 0.129) and grade 6 enjoys PA (*β* = 0.093) and a large direct effect on grade 6 BMI (*β* = 0.943). BMI in grade 5 was the only significant predictor of grade 6 BMI. 

## 4. Discussion

This study tested children’s associations between enjoyment of PA and minutes of MVPA in PE during grade 5 and health relevant outcomes one year later. The purpose of testing these associations was to explore the merits of pursuing these sometimes-competing goals during PE instruction. One of the major strengths of this study is the large and geographically far-reaching sample along with the longitudinal nature of the analysis. Another strength of the study is that it tests whether gaining minutes of MVPA within PE is the mechanism through which experiences lead to BMI- and PA-related outcomes one year later.

One important consideration for this data is that the grade 5 participation in MVPA during physical education class and the minutes in PE class per week were higher in this sample at these time points than they had been two years earlier. In an analysis of grade 3 SECCYD data, Nader et al. [13] found that 37% of PE class time was spent in MVPA, and children participated in 69 min per week of PE. Our estimates exceed these numbers, with 43% of time in PE spent in MVPA and 75 min per week in PE accumulated throughout the week. These trends show both an increase in overall PE time and MVPA proportion within PE in this sample when independent variables were measured. This change in trajectory is important to further understand the relationship of PE for the sample as it had on average been on a trajectory of increasing time and intensity for two years before the measures used in this study.

PE MVPA minutes in grade 5 did not predict sport self-concept, enjoyment of PA, total MVPA minutes, or BMI one year later. PA enjoyment, however, was associated with all of these (except for BMI) one year later. Overall, these findings support the argument that experiences of movement for children in PE classes are critical to future volitional PA participation. In direct alignment with these findings, previous research found that fun and enjoyment in PE result in positive feelings around movement [32], while negative experiences can result in long-term avoidance of physical activities [33]. This is an important potential pitfall for PE instructors and school districts prioritizing the direct provision of minutes of MVPA for established health benefits.

More time in MVPA during PE in grade 5 did not significantly predict grade 5 weekday MVPA. This indicates PE (mean MVPA = 32.57 min/week) was not a major contributor to children’s overall MVPA (mean weekday MVPA = 124.1 min/week) in this sample. The decision to use only weekday MVPA was to be more conservative about the contribution of PE MVPA, as school and thus PE does not take place on the weekends. Even with this conservation, however, PE did not significantly contribute to weekday MVPA. Taken by itself, this insignificant interaction means children are finding more MVPA opportunities outside of PE during the week, and the MVPA in PE is not contributing directly to their finding MVPA opportunities, including weekends, in grade 6. Together, the findings of PE contributing to just under 25% of weekday MVPA along with the accumulation of MVPA during the week contributing to accumulating MVPA one year later suggest a strong approach for PE instruction would be to support PL skills related to finding meaningful PA opportunities outside of school.

Whether the child enjoys PA was a significant predictor variable for weekday MVPA in grade 5 as well as MVPA for all days in grade 6. However, the indirect effect of grade 5 PA enjoyment on grade 6 MVPA through grade 5 MVPA was lower than the direct effect. Taken together, these findings indicate that attaining more MVPA one year later does not only come from attaining more MVPA the year before; instead, enjoyment of PA is a significant contributor in its own right. Other research on exercise behavior has similarly found enjoyment of the experience and supportive interpersonal behaviors from the facilitator of the exercise had direct effects on intention to exercise in the future as well as measured exercise persistence based on gym attendance over a six-month period [34].

Similar findings related to fun and enjoyment have been expressed by children in sporting contexts, where fun is one of the top reasons children report for involvement in sport [35]. The idea of approaching PE with a focus on enjoyable experiences as opposed to achieving MVPA goals is in alignment with PL thinking. In order to have an enjoyable experience, one does not need to also have fun, and one does not always lead to the other. Enjoyment can come from being challenged while remaining focused and relatively in control of an experience—having met basic psychological needs [34]. This philosophy relates to other literatures regarding PE instruction, such as flow state [36] and zone of proximal development [37]. The resulting enjoyment of PA and self-concept has been shown to have positive relational value as well, such as being a mitigating factor for the negative consequences of being cyber bullied [38,39]. Researchers have begun to synthesize a theoretical framework to support the development of student PL through PE, focused on a trained educator creating opportunities for positive movement experiences [40]. The residual variance of sport self-concept had a small correlation with the residual variance of grade 6 enjoys PA. The residual variance of grade 6 MVPA also correlated with sport self-concept though to a weaker degree than enjoys PA. What this indicates is that what does not explain sport self-concept within the model may be similar to what does not explain enjoys PA in the model more so than what does not explain minutes of MVPA in grade 6. Considering all associations in the model, both explained and unexplained variances of sport self-concept are more reliably associated with enjoys PA than minutes of MVPA regardless of where those minutes occur. This was a surprising finding, as one would expect those with higher sport self-concept to be more confident sport participants because they have participated in more MVPA, from factors such as voluntary participation in MVPA during PE, playing sports outside of school, playing sports and games with friends during free time, etc. Given the findings in this model, it seems enjoying PA is the stronger motivator for physical activity self-concept regardless of context.

Within this sample, girls participated in fewer minutes of MVPA in PE class and fewer minutes of MVPA overall, which is consistent with previous literature related to PE [35] and overall MVPA [35,36]. Nader [13] showed from the same sample in grade 3 that boys spent significantly more time in MVPA than girls during PE class (38.3% vs. 35.6% respectively). Surprisingly, limited physical functioning was not associated with any other variable.

In the case of BMI, it is evident that grade 5 BMI was not only a strong predictor of grade 6 BMI but also that grade 5 BMI explained nearly all of the variance for grade 6 BMI. This evidence further supports the findings of Nesbit et al. [12], who showed in this same sample that children tend to stay in their BMI cluster trajectories. They had found MVPA to differ significantly between children with overweight-obese and healthy weight with no difference in sedentary behavior. However, enjoying PA and supplementing more MVPA in PE are both not associated with BMI one year later, and nor is attaining more weekday MPVA overall. Grade 5 BMI having a weak correlation with grade 5 enjoys PA may indicate that some of the significant associations of enjoys PA on outcomes may vary by BMI. There was also a similar relationship of covariance between PE MVPA and BMI, though again PE MVPA did not have any significant associations. Finally, grade 5 BMI had small negative associations with grade 5 weekday MVPA and whether the child enjoyed PA in grade 6. Altogether, these findings evidence that BMI has an effect on self-selecting to participate in MVPA and that attaining more MVPA both during the week and in schools does not significantly affect BMI.

### Limitations

This longitudinal sample had a large attrition rate for measures included in this study. However, FIML estimation is a widely accepted method for handling missing data, having demonstrated unbiased estimates and model fit information as long as data are considered missing at random [41]. The most commonly cited reasons for participants not wearing accelerometers in this dataset were inconvenience and concerns for the appearance around the waist [10]. Previous citations of the accelerometer data show that of those who participated in the longitudinal study at grade 5 (*N* = 1084), 885 agreed to wear the accelerometer, and 850 had valid data; at grade 6 (*N* = 1064), 752 agreed to wear the accelerometer, and 699 had valid data [10]. Retention rates considering participation between these two time points are relatively high, at or above 70%, with only between 35 and 53 participants missing due to improperly wearing the devices. We feel the difference in time point rate of retention (between grades 5 and 6) to constitute missing at random.

Furthermore, this study measures a child’s overall enjoyment of PA from parent report on a 5-point Likert scale. Findings should be understood through the lens that enjoyment of PA in this study is more a measure of disposition than directly influenced by PE classes. The breadth of experiences in PA begins before PE classes and, as shown in these findings, extend beyond the PE class. However, PE classes, like all school subjects, are designed to influence students’ KSAAs (knowledge, skills, attitudes, and abilities), of which enjoyment fits strongly within the “attitudes” category. Finally, these authors consider primary caregiver understanding of whether a child enjoys PA to be a strong source for measurement accuracy as well as potentially less prone to response bias, such as is common in self-report, though more research is warranted to be certain.

## 5. Conclusions

The PL cycle encourages a holistic development of an individual’s relationship to PA by ensuring facilitation of movement confidence, competence, and active participation [22]. This study adds to the evidence of the important opportunity for exercise facilitators to make a difference in future health behavior by thinking about how PA opportunities are delivered to children in a PE context and beyond. The evidence suggests MVPA should be a secondary objective only when it does not contradict with the primary objective: facilitating positive experiences that lead to improvements in confidence, competence, and ultimately enjoyment.

## Figures and Tables

**Figure 1 sports-09-00128-f001:**
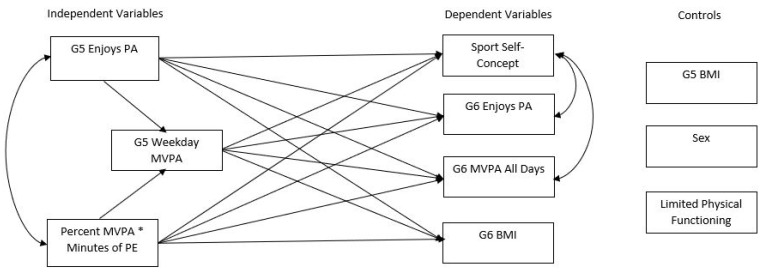
Structural Equation Model Tested Interactions.

**Figure 2 sports-09-00128-f002:**
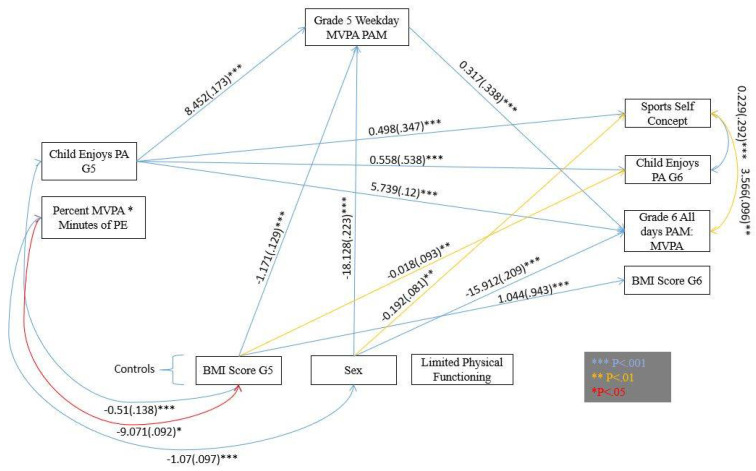
Structural Equation Model Showing Significant Interactions (Unstandardized estimates (standardized estimates) *p*-value category).

**Table 1 sports-09-00128-t001:** Study Variable Descriptions and Sources of Measurement.

Variable		Explanation	Grade
Enjoys PA		Mothers were asked to complete an interview called the “Parent Interview of Child Physical Activity.” This measure is a non-validated questionnaire with 13 items. One item was used for this study: “How much do you agree or disagree with the following statement: My child enjoys participation in physical activity/sports.” The response is a 5-point Likert scale: (1) strongly disagree, (2) somewhat disagree, (3) neutral, (4) somewhat agree, and (5) strongly agree.	5, 6
	SOFIT Percent MVPA	Researchers observed a PE instruction quality of the subjects using the System for Observing Fitness Instruction Time. The SOFIT is considered a more objective observation tool. Percent intervals in MVPA were computed as 100 times the sum of the number of intervals walking and very active divided by the number of intervals observed for cases with complete data.	5
	SOFIT PE Minutes Per Week	During the SOFIT observation, PE teachers were asked how many minutes per week the child participates in PE [29].	5
PE Minutes per Week MVPA		SOFIT percent MVPA was standardized and multiplied with the SOFIT Minutes per week to create an estimate of the minutes per week the subject spent in MVPA during PE classes.	5
	Physical Activity Monitor (PAM)	Computer Science and Applications, Inc. (CSA) single channel accelerometer. Monitors were worn by children for seven consecutive days throughout a typical school week. Average minutes per day of MVPA were calculated as: Moderate (3–5.9 METs) or Vigorous (6–8.9 METs), where: METs = 2.757 + (0.0015 × count) + (−0.08957 × age in years)+(−0.000038 × count × age in years) [12].	5, 6
Weekday MVPA		Total MVPA minutes attained during weekdays (Monday–Friday)	5
MVPA All Days		Total MVPA minutes attained during the entire week	6
Sport Self-Concept		Taken from the “How I Do in School” instrument. Five of the nineteen 7-point likert scale items pertain to sports. Example questions are “How good at sports are you?” and “How well do you expect to do in sports this year?”Mean of the 5 scores were totaled with higher scores, indicating a more positive self-concept of ability in sports. Reliability of the 5 items in the SECCYD dataset is Cronbach’s alpha = 0.85. Items were based on work from Eccles and Wigfield [30].	6
BMI		Subject’s height and weight were measured and recorded by researchers. Measurements were recorded in English standard measurement. In order to reduce measurement error, the English system BMI calculation was used: Weight(lbs)/Height(in)^2^ × 703	5, 6
Sex		Female or male sex—doctor identification at birth.	Enrollment
Limited Physical Functioning		Part of the Child Health Questionnaire completed by the subject’s mother. The questionnaire was designed to measure the mother’s perception of their child’s physical and psychosocial health. Limited Physical Functioning was computed as the mean of 4 items with higher scores indicating more physical limitations due to health problems during the previous 4 weeks [31].	5

**Table 2 sports-09-00128-t002:** Variable Descriptive Statistics.

Variables		N	Mean	SD	Range
Independent Variables, Grade 5					
G5 Enjoys PA		963	4.5	0.827	4
PE Percent MVPA × PE Minutes/Week		875	32.57	22.11	157.24
	PE Percent MVPA	875	0.43	0.18	0.98
	PE Minutes/Week	875	74.63	36.6	281
Indirect Interaction Variable, Grade 5					
Weekday Minutes of MVPA		850	124.1	40.61	313
Dependent Variables, Grade 6					
Sport Self-Concept		1012	5.87	1.19	6
G6 Enjoys PA		900	4.47	0.856	4
Minutes of MVPA All Days		696	96.08	37.86	406.04
BMI		810	20.4	4.95	43.57
Control Variables, Grade 5					
BMI		835	19.64	4.46	33.38
Sex (1 = Male, 51.7%, 2 = Female, 48.3%)		1364	1.48	0.5	1
Limited Physical Functioning		1014	1.15	0.42	2.75

## Data Availability

Data can be obtained by contacting the Inter-university Consortium for Political and Social Research (ICPSR) and completing an application. https://www.icpsr.umich.edu/web/ICPSR/series/00233 (accessed on 10 September 2021).

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
