# Peer review of "Enjoyment of Physical Activity—Not MVPA during Physical Education—Predicts Future MVPA Participation and Sport Self-Concept"

_sports, 2021, doi:10.3390/sports9090128_

Round 1
Reviewer 1 Report
This manuscript well done on a methodological and statistical level deals with a very topical issue: the relationship between motor literacy and physical activity at medium and high intensity, commonly called metabolic. There is a general understanding that physical education is a means of creating a physically healthy population. The authors focus attention on the discordant themes related to primary practices within PE activities for
achieve this end. The main difference is whether PE facilitators should primarily ensure participation children exercise a certain level of energy during the lesson, or develop their confidence, competence, knowledge and motivation for meaningful participation in physical activity (PA) outside the PE class (referred to as physical literacy [PL]). This study uses structural equation modeling for examine associations between enjoyment of Grade 5 PA and minutes of moderate to vigorous physical activity (MVPA) in the PE class and future participation in Grade 6 PA. The results suggest that PE instructors should prioritize PL over improvement and maintenance of high energy expenditure during the PE class for long-term MVPA.
Only small clarifications are needed:
- in addition to the sample inclusion criteria, were there also exclusion criteria?
- was the parental involvement of the families involved considered? It is very important for motivational and enjoyment dynamics (see the recent work that could be cited "Bonavolontà, V., Cataldi, S., Latino, F., Carvutto, R., De Candia, M., Mastrorilli, G.,... Fischetti, F. (2021). The role of parental involvement in youth sport experience: Perceived and desired behavior by male soccer players. International Journal of Environmental Research and Public Health, 18 (16) doi: 10.3390 / ijerph18168698 ";
- Why were variables related to "well-being" not evaluated? These variables with the PE and the extracurricular and enjoyment method can improve equally.
Author Response
Thank you very much for your thorough work reviewing our manuscript. You have very comprehensively and succinctly summarized the main points which is affirming.
For specific responses to points please see the attachment.

Reviewer 2 Report
Congratulations to the authors for their work. It is very interesting in terms of its objectives, sample, methodology used and the applicability of its conclusions.
Some minor changes are recommended to improve the article.
It would be interesting to put the mean age in the abstract, to make it easier for readers from other countries to identify the sample.
In table 1 this sentence is incomplete PE teachers during the SOFIT observation were asked how many minutes per week the subject is enrolled to participate in
Use some reference for System for Observing Fitness Instruction Time (SOFIT)
Use a reference for How I Do in School" instrument.
The discussion can be improved by arguing further for possible causal relationships derived between the variables.
The following quotes are suggested to help.
[1] Benítez-Sillero, J.d.D.; Armada Crespo, J.M.; Ruiz Córdoba, E.; Raya-González, J. Relationship between Amount, Type, Enjoyment of Physical Activity and Physical Education Performance with Cyberbullying in Adolescents. Int. J. Environ. Res. Public Health 2021, 18, 2038. https://doi.org/10.3390/ijerph18042038
[2] Juan D. Benítez-Sillero, Rosario Ortega-Ruiz & Eva M. Romera (2021) Victimization in bullying and cyberbullying and organized physical activity: The mediating effect of physical self-concept in adolescents, European Journal of Developmental Psychology, DOI: 10.1080/17405629.2021.1967136
References not used in the introduction or discussion should not be used in the conclusion without first arguing the relationship between these studies.
A more concrete conclusion after such a process would be appreciated.
Author Response
Thank you very much for your thorough review of the paper and suggestions to make it stronger. Please see attachment for specific feedback to your comments
